# Inositol Restores Appropriate Steroidogenesis in PCOS Ovaries Both In Vitro and In Vivo Experimental Mouse Models

**DOI:** 10.3390/cells13141171

**Published:** 2024-07-09

**Authors:** Valeria Fedeli, Vittorio Unfer, Simona Dinicola, Antonio Simone Laganà, Rita Canipari, Noemi Monti, Alessandro Querqui, Emanuele Galante, Gaia Laurenzi, Mariano Bizzarri

**Affiliations:** 1Department of Experimental Medicine, University La Sapienza, 00185 Rome, Italy; noemi.monti@uniroma1.it (N.M.); alessandro.querqui@uniroma1.it (A.Q.); emanuelegalante3@gmail.com (E.G.); 2Systems Biology Group Lab, University La Sapienza, 00185 Rome, Italy; s.dinicola@lolipharma.it; 3The Experts Group on Inositol in Basic and Clinical Research, and on PCOS (EGOI-PCOS), 00161 Rome, Italy; vunfer@gmail.com; 4Dept. of Gynaecology, UniCamillus—Saint Camillus International University of Health Sciences, 00131 Rome, Italy; 5Unit of Obstetrics and Gynecology, “Paolo Giaccone” Hospital, Department of Health Promotion, Mother and ChildCare, Internal Medicine and Medical Specialties (PROMISE), University of Palermo, 90127 Palermo, Italy; antoniosimone.lagana@unipa.it; 6Department of Anatomy, Histology, Forensic Medicine and Orthopedic, Unit of Histology and Medical Embryology, Sapienza, University of Rome, 00185 Rome, Italy; rita.canipari@fondazione.uniroma1.it (R.C.); gaia.laurenzi@uniroma1.it (G.L.)

**Keywords:** aromatase, polycystic ovary syndrome (PCOS), myo-Inositol, D-chiro-Inositol, steroidogenesis, FSH receptor

## Abstract

Androgen excess is a key feature of several clinical phenotypes of polycystic ovary syndrome (PCOS). However, the presence of FSH receptor (FSHR) and aromatase (CYP19A1) activity responses to physiological endocrine stimuli play a critical role in the pathogenesis of PCOS. Preliminary data suggest that myo-Inositol (myo-Ins) and D-Chiro-Inositol (D-Chiro-Ins) may reactivate CYP19A1 activity. We investigated the steroidogenic pathway of Theca (TCs) and Granulosa cells (GCs) in an experimental model of murine PCOS induced in CD1 mice exposed for 10 weeks to a continuous light regimen. The effect of treatment with different combinations of myo-Ins and D-Chiro-Ins on the expression of *Fshr*, androgenic, and estrogenic enzymes was analyzed by real-time PCR in isolated TCs and GCs and in ovaries isolated from healthy and PCOS mice. Myo-Ins and D-Chiro-Ins, at a ratio of 40:1 at pharmacological and physiological concentrations, positively modulate the steroidogenic activity of TCs and the expression of *Cyp19a1* and *Fshr* in GCs. Moreover, in vivo, inositols (40:1 ratio) significantly increase *Cyp19a1* and *Fshr*. These changes in gene expression are mirrored by modifications in hormone levels in the serum of treated animals. Myo-Ins and D-Chiro-Ins in the 40:1 formula efficiently rescued PCOS features by up-regulating aromatase and FSHR levels while down-regulating androgen excesses produced by TCs.

## 1. Introduction

Polycystic ovary syndrome (PCOS), also known as hyperandrogenic anovulation or Stein-Leventhal syndrome, is a heterogeneous condition that affects from 8% to 20% of women [1]. Currently, according to the Rotterdam diagnostic criteria, PCOS assessment must satisfy two of the following criteria: evidence of clinical and/or biochemical hyperandrogenism, oligo-ovulation and/or anovulation, or confirmation of polycystic ovarian morphology (string of pearls) at ultrasound analysis (‘string of pearls’ appearance) [2]. However, several co-morbidities, including insulin resistance [3] and metabolic syndrome [4], may complicate the clinical picture, promoting the emergence of different pathophenotypes. Diagnostic assessment of PCOS according to such criteria has been the subject of extensive criticism, given that the Rotterdam-based approach does not capture several clinical aspects, leaving room for unsupported interpretations [5]. In addition, using the Rotterdam criteria does not help in searching for a personalized treatment. Microbiota deregulation [6], genetic factors [7], and low-grade inflammatory status [8] can also contribute to adding further complexity to the disease.

This multifaceted nature of PCOS makes it a challenging condition to manage and study, with ongoing research aimed at better understanding its underlying mechanisms to identify clinical phenotypes amenable to tailored therapies [9].

According to Rotterdam’s criteria, hyperandrogenism (HA) is a common feature of PCOS phenotypes [2]. Several studies demonstrate that the ovary chiefly provides a sustained increase in androgens, principally testosterone and DHEA [10]. Besides the central role played by the hypothalamic-pituitary axis in promoting ovarian androgen synthesis through gonadotropins, some autocrine and paracrine factors exert an additional function in modulating and enhancing the ovarian synthesis of androgens [11]. Compelling evidence has confirmed that the increased androgen secretion is a hallmark of Teca cells (TCs) in PCOS ovaries [10,12].

However, the increased release of androgens does not explain all the observed molecular and clinical characteristics of PCOS. Little attention has been paid to the concomitant changes occurring in FSH receptor (FSHR) availability and estrogen synthesis. The appropriate expression of aromatase (CYP19A1) in the ovary is crucial for the autocrine regulation of folliculogenesis, the endocrine control of the female reproductive tract, and the coordination of gonadotropin release [13]. Some of the oldest studies contended that a hypothetical mechanistic explanation of PCOS could be ascribed to a condition of «ineffective» estrogen competency due to decreased activity of CYP19A1 and FSHR. Inadequate availability of both estradiol and FSHR leads to an «estrogen insufficiency» and consequent arrest of follicle growth [14,15]. Noticeably, in PCOS ovaries, the persistent stimulation exerted by gonadotropins (LH) increases the androgen release from TCs. In contrast, Granulosa cells (GCs) are incapable of forming adequate amounts of estrogen, primarily due to LH-induced inhibition of *Cyp19a1* expression [16,17]. Additionally, CYP19A1-deficient girls develop ovarian cysts similar to those found in polycystic ovarian syndrome [18]. Lack of 17β-estradiol release in due time and at appropriate levels can ultimately determine the impairment of folliculogenesis. This hypothesis has been vindicated by a recent study demonstrating that a PCOS-like syndrome can be induced in mice by treating animals with Letrozole, a well-known CYP19A1 inhibitor [19,20].

Recent studies have demonstrated that myo-Inositol (myo-Ins), associated with low concentrations of D-Chiro-Ins, can effectively rescue ovary function in women affected by PCOS (reviewed in [21]). Specifically, women affected by PCOS phenotypes displaying HA upon inositol treatment show a significant improvement in clinical and biochemical markers, while in patients with PCOS without androgen involvement, benefits were only marginal [22]. These findings agree with a compelling body of evidence demonstrating that inositol can modulate ovarian steroidogenesis, reviewed in [23], namely by counteracting clinical signs of hyperandrogenism [24,25]. However, studies investigating the molecular mechanisms and factors involved in inositol modulation of ovarian steroidogenesis are still lacking.

To manage such issues, we investigated the usefulness of myo-Ins and D-Chiro-Inositol (D-Chiro-Ins) in a light-induced PCOS model to verify the model’s reliability and the potential benefit of inositol-based treatments. Rodent models are powerful tools for investigating several dynamic conditions that can trigger a PCOS-like condition in experimental animals. Hormonally induced PCOS models are among the most investigated [26,27], while in recent years, models based upon genetic manipulation have also attracted attention [28]. A less common but intriguing mouse model is the light-induced PCOS model [29]. In this model, mice were exposed to continuously illuminated conditions 24 h a day (constant light) instead of the typical 12 h light-dark cycle. It is worth noting that the disruption of the mouse circadian rhythm can lead to hormonal and metabolic disturbances similar to those observed in women with PCOS, including hyperandrogenism and insulin resistance [30]. Specifically, this model permits the avoidance of sustained androgen administration while reproducing the most relevant PCOS features, including hyperandrogenism, insulin resistance, and ovarian/reproductive dysfunctions [31].

We hypothesize that down-regulation of aromatase and FSH receptors represents a main pathogenic event in PCOS. Therefore, myo-Ins and D-chiro-Ins in the appropriate ratio (40:1) could likely restore a proper steroidogenic pattern by modulating androgenic enzymes while increasing the synthesis of CYP19A1 and FSHR.

## 2. Materials and Methods

### 2.1. Mouse Model of PCOS

Thirty-day-old CD1 female mice were used. Animals were assigned to either the experimental arm (experimental induction of PCOS; *n* = 20) or the control group (healthy; *n* = 10). PCOS was induced by placing mice in routine housing conditions, except for the light cycle that was extended to 24 h (constant light, L/L) for 10 weeks, as previously described [30]. The control group of mice was kept under normal light and dark conditions. The animals were weighed at 1 month of age and at the end of the 10 weeks of PCOS induction treatment. The data were compared with the respective healthy controls. Moreover, the ovaries isolated from healthy and PCOS-induced animals were cleared of surrounding tissues and weighed. Blood was collected to analyze serum levels of DHEA, progesterone, and estrogen in the different experimental conditions. All animal procedures have been approved by the Sapienza University ethics committee for animal research and conform to the International Guiding Principles for Biomedical Research Involving Animals guidelines as recommended by the World Health Organization for the use of laboratory animals, as well as the ARRIVE guidelines validated by the Enhancing the Quality and Transparency of Health Research (EQUATOR) Network.

### 2.2. Primary Cultures of Granulosa and Theca Cells and Inositol and D-Chiro-Inositol Treatments

HEALTHY (*n* = 30) and PCOS-induced (*n* = 48 + 30) mice were subcutaneously injected with 10 I.U. Pregnant Mare Serum Gonadotropin (PMSG, Folligon, Intervet, Milan, Italy) to stimulate folliculogenesis. After 46 h of PMSG treatment, animals were sacrificed using CO_2_, and ovaries were collected.

GCs and TCs were isolated from ovaries following the protocol previously described [32], and primary cultures (*n* = 3) were seeded into 60 mm dishes (BD Falcon) in DMEM-High Glucose supplemented with 5% fetal bovine serum (FBS), 100 IU/mL penicillin, 100 μg/mL streptomycin, 0.4 mg/mL Amphotericin B, and Gentamicin. Cell cultures were kept at 37 °C in an atmosphere of 5% CO_2_ in air.

When cells reached sub confluence, culture medium was substituted with DMEM-Low Glucose without FBS and supplemented with 100 IU/mL penicillin, 100 μg/mL streptomycin, 0.4 mg/mL Amphotericin B, and Gentamicin. The shift from high to low glucose is required to improve inositol uptake [33]. Two hours after plating, cells were treated for 10 min with 10 ng/mL LH (R&D System; Bio-Techne SRL, Milano, Italy) and 100 ng/mL Insulin to activate the final phases of functional maturation and steroidogenic activity. Cells were then treated for 90 min with myo-Ins and D-Chiro-Ins (kindly provided by Lo. Li. Pharma s.r.l., Rome, Italy) in the continuous presence of LH and Insulin. At the end of culture, cells were collected, and total RNA was extracted.

Sigma-Aldrich (St. Louis, MO, USA) supplied all reagents for cell cultures unless otherwise specified.

### 2.3. Vivo Myo-Inositol and D-Chiro-Inositol Treatments

After 10 weeks of PCOS induction, healthy (*n* = 10) and PCOS (*n* = 20) animals were treated in vivo with myo-Ins: D-Chiro-Ins 40:1 ratio at a concentration of myo-Ins 0.5 mM and D-chiro-Ins 12.5 µM. The treatment with inositols was through water; all mice had personal bottles. On average, each mouse drinks 5 mL of water per day. Every morning, 5 mL of water supplemented with myo-Ins and D-Chiro-Ins were inserted into the bottle (additional water was eventually provided if the bottles were empty). For each animal, the total water consumed and body weight were noted daily. At the end of the 10-day treatment, ovaries and adipose tissue were collected for total RNA extraction. Blood was collected for hormonal evaluation.

### 2.4. RNA Isolation, RT- and Real-Time-PCR Analyses

Total RNA was isolated from TCs, GCs, and whole ovaries with TRI Reagent^®^ (Sigma-Aldrich), according to the manufacturer’s protocol. The concentration of RNA was determined with Nanodrop-ND 2000 (Thermo-Fisher Scientific, Waltham, MA, USA). Each sample was reverse transcribed using the Optifast cDNA synthesis kit (Aurogene S.r.l., Roma, Italy) according to the manufacturer’s protocol. Real-time PCR was performed with SYBR Green qPCR Mastermix (Bio-Rad Laboratories S.r.l., Segrate (MI), Italy) using a Bio-Rad CFX384 Touch Real-Time PCR. The sequences of the used primers are reported in Table 1. Each sample was normalized to its ribosomal protein S29 (Rps29) content [34], and the ΔΔCt method was used to determine the levels of gene expression.

### 2.5. Testosterone, Estrogen, DHEA, and Progesterone Assay

Testosterone, estrogen, DHEA, and progesterone serum levels were evaluated by ELISA with MyBioSource ELISA kits (Aurogene S.r.l.) according to the manufacturer’s protocol. Tests were performed on blood collected from healthy (*n* = 5) and PCOS-induced (*n* = 5) animals after 10 days of in vivo treatment with myo-Ins and D-Chiro-Ins.

### 2.6. Statistical Analysis

Statistical analysis was conducted using one-way analysis of variance (ANOVA), followed by the Bonferroni test for comparisons of multiple groups or a two-tailed *t*-test when comparing data derived from two groups. Values with *p* ≤ 0.05 were considered statistically significant. Statistical analysis was performed with GraphPad 8.01 Instat software (GraphPad Software, Inc.; San Diego, CA, USA).

## 3. Results

### 3.1. In the Experimental Mouse Model of PCOS, the Continuous Light Exposure Induces Overweight

When the light/dark daily cycle is perturbed by exposure to continuous light for almost 20–25 days, female rodents progressively develop chronic anovulation and other PCOS features (including irregular cycles, oligo-anovulation, polycystic ovaries, and follicle atresia). This is a reversible condition, given that most animals recover a normal estrous cycle when switched to a normal light/dark regimen [35]. Albeit some unavoidable limitations, rodent PCOS models based upon the disruption of the light/dark cycle provide a valuable alternative to models based on pharmacological interventions (androgen/estrogen administration, CYP19A1 inhibition), namely by avoiding off-target effects of hormone inducers [36,37].

According to a previously reported protocol [38], we induced a PCOS-like syndrome by housing mice in continuous light for 10 weeks. As a control group (healthy), we kept 10 mice under normal light and dark conditions. Animals were weighed weekly during the conditioning period.

In the PCOS-induced mice, weight increases on average from 19.23 ± 0.73 g to 31.24 ± 2.48 g at the end of the 70-day observation time (Figure 1a). Notably, 5% of mice developed obesity, with increments averaging twice the weight of PCOS-induced animals, reaching 62.05 ± 3.67 g. Control healthy mice show only a mild increase, from 19.54 ± 1.23 g to 25.16 ± 1.35 g (Figure 1a). Similarly, ovaries from PCOS animals show a significant weight increase with respect to control mice: 0.014 g ± 0.0024 vs. 0.0062 ± 0.0016 (Figure 1b).

### 3.2. The Continuous Light Exposure Increases the Expression of Genes in the Steroidogenic Pathway

To ascertain if continuous light exposure could modify the endocrine pattern in PCOS-induced animals, we investigated the mRNA levels of critical enzymes belonging to the steroidogenic pathway in ovaries obtained from healthy and PCOS-induced animals. In ovaries obtained from PCOS mice, we observed a significant increase in mRNA transcripts related to *Cyp17a1*, *Hsd3b*, and *Hsd17b*, genes involved in the synthesis of DHEA, androstenedione, and testosterone (Figure 2a–c).

All the mRNA for these enzymes increased in PCOS mice. Noticeably, *Hsd17b*-mRNA levels displayed a twenty-fold increase over controls. Since HSD17β in TCs is thought to be the key androgenic enzyme regulating testosterone release [39], the natural precursor that would be transformed into estradiol by CYP19A1 expressed by GCs, we verified if such an increase could be associated with a concomitant systemic effect. We analyzed serum levels of testosterone in healthy and PCOS animals. As expected, we ascertained a significant increase in serum testosterone in PCOS mice (Figure 2d).

Moreover, we analyzed the expression of mRNA transcripts of two mRNAs specific to GCs, *Cyp19a1* and *Fshr*. In agreement with our working hypothesis, we observed that the expression of *Cyp19a1* was statistically downregulated in PCOS ovaries. Conversely, no difference was found in *Fshr*-mRNA expression (Figure 3).

Overall, these results suggest that the experimental PCOS model promoted a PCOS-like condition, namely by fulfilling the basic required features, as represented by an androgenic Theca phenotype (with increased testosterone levels). In contrast, in GCs, the expression of *Cyp19a1* was drastically reduced.

### 3.3. In Vitro Treatment of Theca and Granulosa Cells with Pharmacological Concentrations of Myo-Ins and D-Chiro-Ins

To identify the more effective molar ratio at which the two inositols, administered in a pharmacological dose (>1 mM), displayed the most significant effect, we isolated GCs and TCs from healthy (*n* = 12) and PCOS (*n* = 18) mice as described in the Methods. To validate the TC and GC purity, we evaluated by real-time PCR the presence of mRNA for *Fshr* and *Cyp191a*, two specific markers of GCs, and *Cyp17a1* and *Hsd3b*, two specific markers of TCs. As expected, the expression of *Fshr*- and *Cyp19a1*-mRNA were mainly observed in GCs, while in TCs, they were almost undetectable; on the contrary, we recorded the expression of *Cyp17a1* and *Hsd3b*, almost only in TC [32].

The cells were treated for 90 min in the continuous presence of LH/insulin, with myo-Ins and D-Chiro-Ins at different ratios, as shown in Table 2. LH/insulin alone was used as a positive control. The choice of different ratios of inositols is justified in light of previous studies demonstrating that in vivo treatment of PCOS mice with different myo-Ins and D-Chiro-Ins ratios provided significantly different histological and clinical outcomes. Bevilacqua et al. showed that PCOS features could be reversed only when myo-Ins/D-Chiro-Ins were added in the 40:1 ratio, while other formulas were either ineffective or worsened the clinical picture [38].

Stimulation with LH/Insulin showed a trend toward a downregulation of the expression of *Hsd17b*- and *Cyp17a1*-mRNA. Cotreatment with all different myo-Ins/D-Chiro-Ins concentrations caused a further decrease. However, values did not reach statistical significance (Figure 4a,b).

Conversely, LH/Insulin stimulation triggers an increase in the expression of *Hsd3b*-mRNA, even though it is not statistically significant (Figure 4c). HSD3B is the key enzyme responsible for the conversion of DHEA into androstenedione and then into testosterone [40]. This result agrees with a previous report indicating that insulin promotes androgenic pathways mainly through the upregulation of *Hsd3b* in TCs [41]. It is worth noting that myo-Ins/D-Chiro-Ins efficiently counteracts the increase in *Hsd3b*-mRNA after LH/Insulin stimulation in the samples treated with myo-Ins/D-Chiro-Ins at a ratio of 20:1 and 40:1 (Figure 4c).

Conclusively, myo-Ins/D-Chiro-Ins modulates the expression of androgenic enzymes, specifically by significantly downregulating the expression of *Hsd3b*.

We treated GCs under the same conditions as TCs. In GCs obtained from healthy animals, the LH/Insulin stimulation slightly increases *Cyp19a1* expression. In contrast, after the addition of myo-Ins and D-Chiro-Ins, a significant reduction in *Cyp19a1* expression can be observed, but only in the cells treated with the highest myo-Ins/D-Chiro-Ins ratio concentration (5:1) (Figure 5a). On the contrary, in PCOS-GCs, LH/Insulin stimulation does not trigger a significant increase in *Cyp19a1*-mRNA. Similarly, treatment with inositols does not induce appreciable changes, with the very relevant exception of samples treated with myo-Ins and D-Chiro-Ins at the 40:1 ratio. In this case, a three-fold increase in *Cyp19a1* expression was observed (Figure 5b).

Furthermore, we investigated mRNA expression for *Fshr* following in vitro treatment of GCs obtained from healthy and PCOS animals. As shown in Figure 5c, cells obtained from healthy mice responded to LH/insulin stimulation by increasing *Fshr*-mRNA. The stimulation with the different doses of myo-Ins and D-Chiro-Ins counteracted this increase. On the contrary, GCs obtained from PCOS mice did not respond to LH/Insulin treatment (Figure 5d). However, we observed a significant increase in *Fshr*-mRNA expression in the presence of myo-Ins and D-Chiro-Ins at the 40:1 ratio (Figure 5d).

Treatment of GCs from PCOS ovaries with myo-Ins/D-Chiro-Ins at the 40:1 ratio was highly effective in restoring *Cyp19a1* and *Fshr* expression, two factors that are mandatory to ensure estrogen synthesis and proper hormonal feedback to the hypothalamus-pituitary-gonadal axis.

### 3.4. Treatment of Theca and Granulosa Cells of PCOS Mice with Physiological Concentrations of Inositols

To understand if myo-Ins could also exert a relevant role within a physiological range of concentrations, once it was established that the 40:1 ratio is the more appropriate ratio for allowing normal steroidogenic activity, we decided to investigate a panel of different sub-pharmacological concentrations of myo-Ins and D-Chiro-Ins at the 40:1 ratio.

We evaluated the expression of several genes involved in the steroidogenesis pathway in TCs (*Cyp17a1* and *Hsd3b*) and GCs (*Cyp19a1*, *Fshr*, and *Hsd17b*) by using five different physiological concentrations of myo-Ins and D-Chiro-Ins, as reported in Table 3. Myo-Ins 1 mM and D-Chiro-Ins 25 µM represent the pharmacological concentrations. The physiological concentrations of myo-Ins (ranging from 0.005 to 0.5 mM) reported in the panel constitute different myo-Ins levels usually measured in a wide range of mammalian cells, although the 0.5 mM for myo-Ins and 12.5 µM for D-Chiro-Ins should be appropriately considered a «supra-physiological» (i.e., integrative) concentration, as those generally used in supportive nutritional preparations [42].

TCs and GCs, isolated from PCOS mice as described in the Methods, were treated for 90 min in the continuous presence of LH/insulin with myo-Ins/D-Chiro-Ins (at the 40:1 ratio) at the concentrations indicated in Table 3.

Both *Cyp17a1*- and *Hsd3b*-mRNA transcripts show a similar pattern of response. At low concentrations of myo-Ins (<0.01 mM), the expression resulted unaffected, while with higher concentrations, we observed a paradoxical increase (Figure 6a,b). We observed the maximal stimulation of both mRNAs at 0.05 mM/1.25 µM myo-Ins and D-Chiro-Ins and a significant decrease at higher concentrations.

Treating GCs with inositols at physiological concentrations increased *Cyp19a1*, *Hsd17b*, and *Fshr* mRNA values. This increase becomes statistically significant for *Cyp19a1* and *Hsd17b* at supra-physiological concentrations of inositols (0.5 mM and 12.5 µM for myo-Ins and D-Chiro-Ins, respectively) (Figure 7a,b). The expression of *Fshr* mRNA was significantly promoted up to three-fold the basal value, even at the lowest inositol(s) concentrations (0.005 M and 0.125 mM for myo-Ins and D-Chiro-Ins, respectively) (Figure 7c).

Overall, those data demonstrated that inositols in supra-physiological concentrations are mandatory to maintain an appropriate androgen synthesis in TCs, thus providing an adequate amount of the endocrine precursor (i.e., testosterone) that, in turn, will be transformed into estradiol by GCs upon the concerted interaction of FSH and its receptor and CYP19A1 activity.

### 3.5. Effects on Ovarian Steroidogenic Enzyme Expression after In Vivo Treatment with Inositols

Once it was established from in vitro experiments that the 40:1 myo-Ins/D-Chiro-Ins ratio is the most effective in restoring steroidogenic activity in ovarian cells, we decided to explore if this treatment could also be effective in vivo. We enrolled 40 mice, 15 of whom were assigned to the control group (healthy), and 25 animals were exposed to a continuous light cycle for 10 weeks (PCOS), as previously described, to induce PCOS. After the 10 week induction of PCOS, both healthy and PCOS animals were randomly distributed into 4 groups: 2 control groups (healthy and PCOS not treated, *n* = 5) and 2 treated groups in which healthy (*n* = 10) and PCOS (*n* = 20) mice were treated with a supra-physiological concentration of myo-Ins (0.5 mM) and D-chiro-Ins (12.5 µM), according to the 40:1 ratio. Inositols were added to the water, and treatment lasted 10 days. During this period, the continuous exposure to light was upheld to be sure that the positive effect on the body, hormones, and molecular pathways was due to inositol and not to the return of the normal cycle of light and darkness.

Ovaries were isolated from healthy and PCOS-inositol-treated and not-treated animals, and total RNA was extracted. The expression of *Cyp17a1* and *Hsd3b* as markers of TC activity and *Cyp19a1* and *Fshr* as markers of GC endocrine activity was evaluated by RT-PCR. As expected, both *Cyp17a1* and *Hsd3b* are constitutively increased in PCOS animals and do not show any significant change upon inositol treatment (Figure 8a,b). Conversely, in ovaries from healthy mice, inositol stimulation highly enhanced both enzymes (Figure 8a,b).

Concerning genes expressed by GCs, as expected, both *Cyp19a1* and *Fshr* are expressed at significantly lower levels in PCOS animals. Inositol treatment was ineffective in healthy animals, while in PCOS mice, it increased their expression and almost completely restored the synthesis of both mRNAs, thus allowing the ovary to recover a physiological endocrine profile (Figure 8c,d).

We also evaluated the mRNA levels of *Hsd17b*, an enzyme involved in androgen synthesis in the TCs, and estrone conversion into estradiol within the GCs [43]. The expression levels are significantly upregulated in PCOS ovaries, and inositol does not significantly modify this picture. Similarly, inositol does not exert an appreciable effect on the ovaries of healthy mice (Figure 8e).

These data are difficult to interpret, given that both TCs and GCs synthesize the enzyme, and, therefore, we cannot exclude a differential effect displayed by inositols on the two cell types in PCOS mice. Indeed, we observed that inositol treatment was able to increase *Hsd17b* expression in GCs from PCOS animals (Figure 7b), thus facilitating the estrone conversion into 17b-Estradiol.

### 3.6. In Vivo Treatment with Inositols Inhibits Hsd17b Gene Expression in Adipose Tissue

In PCOS women, changes involving the activity and the expression of steroidogenic genes are not limited to the ovary and may likely involve other tissues [44]. Specifically, an increase in the expression of enzymes involved in androgen production, such as the *Hsd17b* gene, can contribute to elevating the overall testosterone release in the general circulation, thus worsening the androgenic phenotype associated with PCOS. We tested this hypothesis by examining the gene expression of *Hsd17b* in the adipose tissues of both healthy and PCOS-induced animals treated with myo-Ins/D-Chiro-Ins at a 40:1 ratio. As depicted in Figure 9, we found that *Hsd17b* levels dramatically increased in PCOS animals, up to 25-fold compared to healthy controls, and inositol treatment was able to reduce such an increase in a very significant manner (~75% of downregulation). A similar trend, albeit not significant, was also observed in adipose tissue obtained from healthy animals. This result suggests that inositol therapy decreases *Hsd17b* expression in adipose tissues and efficiently counteracts the systemic androgenic phenotype associated with PCOS.

### 3.7. In Vivo Treatment with Inositols Reduces the Levels of Circulating Androgenic Hormones

To further confirm these results, we assessed the concentrations of circulating steroid hormones in PCOS animals, to verify if inositol treatment could modify the endocrine pattern in these conditions. Therefore, we collected blood from animals once sacrificed and then tested it through an ELISA assay.

We measured serum levels of DHEA, estrogen, the main product of a fully active steroidogenic pathway, and progesterone in PCOS animals, treated or not treated for 10 days with supra-physiological concentrations of inositols (0.5 mM and 12.5 µM for myo-Ins and D-Chiro-Ins, respectively), in the 40:1 ratio. Data were compared to control PCOS mice not exposed to inositol treatment. DHEA, the primary precursor produced in the androgenic pathway, was significantly reduced upon inositol treatment (Figure 10a). Conversely, estradiol and progesterone levels were significantly upregulated (Figure 10b,c). Progesterone is one of the key hormones in the female endocrine system and is essential for regulating the menstrual cycle, pregnancy, and other physiological functions. Low production of progesterone can have consequences for fertility and health. Overall, those findings indicate that inositols favor the release of endocrine factors involved in fertility and menstrual control while reducing the levels of circulating androgenic hormones.

## 4. Discussion

The experimental model that enacts the appearance of a PCOS-like syndrome in mice, obtained through continuous exposure to light for 70 days, induces significant changes in body weight, which are usually associated with PCOS development in animal models [45] and the ovarian steroidogenic pathway. Namely, ovaries isolated from PCOS animals showed that androgenic enzymes *Cyp17a1*, *Hsd3b*, and *Hsd17b* are highly expressed. They support the emergence of an androgenic systemic phenotype, as confirmed by the increase in serum testosterone levels (Figure 2d). At the same time, ovaries from PCOS mice showed a severe reduction in *Cyp19a1* levels (Figure 3a). This critical enzyme terminates the conversion of androgens into estrogens in GCs. Furthermore, adipose tissues from PCOS mice showed an increased expression of the androgenic enzyme *Hsd17b*. These results confirm that a critical imbalance arises in the steroidogenic pathway in the ovary of our PCOS-induced animals, leading to increased androgen release and reduced availability of estrogens, primarily due to CYP19A1 impairment.

Data provided by in vitro studies, in which isolated TCs and GCs have been treated with myo-Ins and D-Chiro-Ins at different ratios (5:1, 20:1, and 40:1), at pharmacological concentrations (with myo-Ins ≥ 1 mM), showed that inositols inhibited several steroidogenic genes in TCs from PCOS mice at all ratios utilized. On the contrary, *Cyp19a1* expression in PCOS cells is only marginally enhanced by stimulation with LH or Insulin. At the same time, adding myo-Ins/D-Chiro-Ins at the 40:1 ratio significantly increased *Cyp19a1* and *Fshr* expression, thus counteracting one of the main features of PCOS syndrome. These effects are not surprising, as this is the physiological ratio displayed by myo-Ins and D-Chiro-Ins in biological fluids [46], and several studies have already highlighted that this is the more suitable combination, as vindicated by experimental [38] and clinical work (reviewed in [47]).

These findings suggest that in resting conditions, GCs from PCOS mice show poor responsiveness to physiological signals aimed at increasing CYP19A1 and FSHR. Conversely, upon inositol treatment, the ovaries recover a proper estrogenic profile. Overall, these data provide strong confirmation that myo-Ins and D-Chiro-Ins in the ratio of 40:1 at pharmacological concentrations down-regulate some androgenic enzymes (HSD3B) while re-establishing a physiological response of CYP19A1 and FSHR, thus overcoming the main steroidogenic abnormalities found in PCOS.

Studies in vitro in which myo-Ins and D-Chiro-Ins were added at physiological concentrations, albeit at the same 40:1 ratio, provided some different results that can help in understanding the physiological role played by inositol. Inositols are required to ensure androgen synthesis in TCs to ultimately synthesize testosterone, which is further transformed into estrogens by GCs.

The expression of *Cyp17a1* and *Hsd3b* is unaffected below a threshold value (0.01 mM for myo-Ins and 0.25 µM for D-Chiro-Ins); they are significantly stimulated at 0.01 mM for myo-Ins and 0.25 µM for D-Chiro-Ins; however, at higher concentrations, we observed a significant decrease in stimulation. Similar results were obtained with GCs; inositols below the same threshold value have little impact upon the expression of *Cyp19a1* and *Hsd17b,* while for higher inositol concentrations, the expression of the enzymes increases up to several folds. However, at the highest concentration, the stimulation was less evident. Notably, the induction of *Fshr* displays remarkable sensitivity, even in the presence of the lowest inositol concentrations, suggesting that inositols are mandatory for enacting the activation of the *Fshr* gene. This result helps in understanding why ovarian stimulation in women committed to assisted reproductive technologies (ART) is highly facilitated by myo-Ins treatment, which can increase the sensitivity of polycystic ovaries to gonadotropins, leading to a reduction in the doses of FSH required to induce ovulation [48]. However, inositols can be detrimental at higher concentrations, especially when *Cyp19a1* expression is considered. Indeed, increasing inositol concentrations resulted in a significantly lower expression of the genes we evaluated. These data suggest that after a threshold concentration, inositols eventually promote a severe impairment of the ovarian tissue and functions, as has already been noticed [49,50].

Conclusively, inositols in physiological concentrations positively modulate the steroidogenic activity of TCs and significantly increase the GCs’ endocrine competence by increasing the expression of *Cyp19a1* and *Fshr*. It is worth noting that *Cyp19a1* impairment seems to be a critical hub in PCOS pathogenesis, as reported by a recent study performed on rodent models based on continuous light exposure and in human PCOS ovarian cells [51].

Experiments carried out in vivo provided substantial confirmation of the abovementioned results. Supra-physiological concentrations of inositols at the myo-Ins/D-Chiro-Ins 40:1 ratio statistically stimulate expression in healthy animals. At the same time, modify only marginally the expression pattern of the same enzymes in PCOS mice (Figure 8). Conversely, inositols significantly increased *Cyp19a1* and *Fshr* expression in PCOS animals compared to untreated PCOS controls.

A possible explanation is that *Cyp17a1* and *Hsd3b* are already highly expressed in PCOS animals compared to the healthy ones, while the opposite is true for *Cyp19a1* and *Fshr* expression. Therefore, in the first case, inositols cannot further stimulate androgenic enzyme expression. In contrast, inositols can stimulate the expression of *Cyp19a1* and *Fshr*, genes that are downregulated in PCOS mice. These changes in gene expression are mirrored by the complementary modifications observed in hormone values assessed in the blood of treated animals. Indeed, because of the rescued granulosa steroidogenic function, mice in vitro treated with inositols at the 40:1 ratio showed increased levels of estrogen and progesterone, while DHEA significantly decreased.

The fact that inositol does not significantly impair the androgenic activity of the ovary cannot be viewed as a negative result. Indeed, TCs must continue releasing androgen precursors to fuel the subsequent conversion into estrogens performed in the granulosa layer. A physiological «recuperation» of the ovarian estrogenic function cannot be completed without an appropriate synthesis of androgen precursors. Human adipose tissue is capable of active androgen synthesis catalyzed by HSD17B, and increased expression in obesity may contribute to circulating androgen excess in PCOS women [52]. The significant reduction in the synthesis of *Hsd17b* within the adipose tissue suggests that ovarian steroidogenic deregulation may influence in some way the peripheral metabolism of androgens, paving the way for the emergence of an «androgenic phenotype». On the contrary, our findings advocate that inositols efficiently counteract some systemic effects of PCOS upon metabolism and peripheral synthesis of androgens.

The molecular studies we performed provided a mechanistic explanation of the molecular activity exerted by inositol in the ovary while suggesting a reliable rationale for the use of myo-Ins and D-Chiro-Ins in PCOS treatment, as reported by several clinical studies and randomized trials [53,54]. A recent meta-analysis performed on twenty-six randomized clinical trials (including data from 1691 patients) has shown that myo-Ins significantly reduces androgens in PCOS women while normalizing cell cycle length, displaying a superimposable clinical efficiency to metformin without any significant side effects [55]. Furthermore, myo-Ins is also effective in normalizing ovarian function and improving oocyte and embryo quality in PCOS women who undergo assisted reproductive treatments, namely by increasing the ovary sensitivity to endocrine stimulation and reducing the doses of FSH administered to induce ovulation [56,57].

However, several questions remain open. First, animal models are priceless; however, they can hardly capture and recapitulate all the complexity behind PCOS. Studies on humans are indeed mandatory. Although numerous evidence suggests that inositols rescued the estrogenic function in PCOS women [57,58,59], a direct study performed on TC, and GCs obtained from women is still lacking. Second, understanding the molecular mechanisms triggered by inositol administration is still in its infancy. Namely, the exact role played by D-Chiro-Ins awaits a compelling analysis. First, Nestler et al. [60] suggested that D-Chiro-Ins could efficiently be used in the management of PCOS because administration of this inositol proved to significantly increase the action of insulin in patients with PCOS, thereby improving ovulatory function and decreasing serum androgen concentrations, blood pressure, and plasma triglyceride concentrations. However, later studies revealed that increasing D-Chiro-Ins concentrations (up to 2400 mg/day) leads to opposite results, with increased testosterone levels and impaired ovarian function [61]. Indeed, it was further reported that the physiological myo-Ins and D-Chiro-Ins ratio in the ovary is severely altered, with myo-Ins being prevalently epimerized into D-Chiro-Ins [62]. The myo-Ins and D-Chiro-Ins imbalance in the ovarian tissue can be a significant causative factor in disrupting the steroidogenic ovary function given that high D-Chiro-Ins concentrations favor testosterone synthesis [63] and promote hypertrophy/hyperplasia of the Theca layer with detrimental effects upon Granulosa activity, as demonstrated in an experimental model of PCOS [38]. We have already proposed [64] that D-Chiro-Ins and myo-Ins exert opposite control over ovary steroidogenesis, specifically on estrogen synthesis through CYP19A1 and FSHR availability. While D-Chiro-Ins inhibits estrogen production, as observed in male subjects [65], myo-Ins promotes CYP19A1 and FSHR activity, as reported in the present study. It is noteworthy that this effect emerges even at physiological inositol concentrations, thus highlighting that the inositol-mediated effect could be considered a primary, physiological effect upon steroidogenesis.

We posit that restoring D-Chiro-Ins levels can be beneficial in ameliorating insulin resistance in non-ovarian tissues, thereby reducing the insulin stimulation upon TCs and, hence, the consequent insulin-dependent enhancement of androgenesis. However, D-Chiro-Ins administration exceeding the need to mitigate insulin secretion would likely impair ovary function by dramatically decreasing myo-Ins levels in the ovary and, consequently, by inhibiting both CYP19A1 and FSHR availability. As a logical consequence, D-Chiro-Ins administration should be limited to those patients in whom PCOS is associated with insulin resistance and concomitant metabolic syndrome. The authors should discuss the results and how they can be interpreted from the perspective of previous studies and the working hypotheses. The findings and their implications should be discussed in the broadest context possible. Future research directions may also be highlighted.

## 5. Conclusions

In conclusion, myo-Inositol exerts priceless regulatory activities upon ovary steroidogenesis and has been shown to efficiently rescue PCOS features by upregulating *Cyp19a1* and *Fshr* levels while downregulating androgen excesses produced from TCs (Figure 11).

## Figures and Tables

**Figure 1 cells-13-01171-f001:**
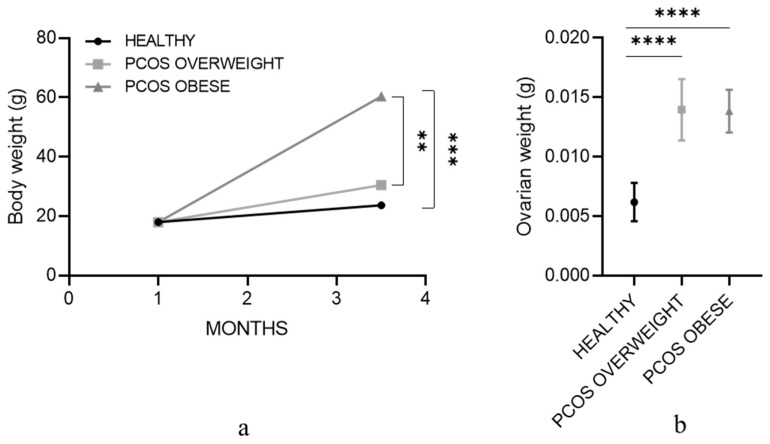
Effect of PCOS-induction on body and ovarian weight. Mean of (**a**) mouse body and (**b**) whole ovary weight at the beginning and after 70 days of treatment, with a total animal number of 10 controls (healthy) and 20 PCOS. Values are represented as the mean ± SEM. ** *p* < 0.01, *** *p* < 0.001, **** *p* < 0.0001.

**Figure 2 cells-13-01171-f002:**
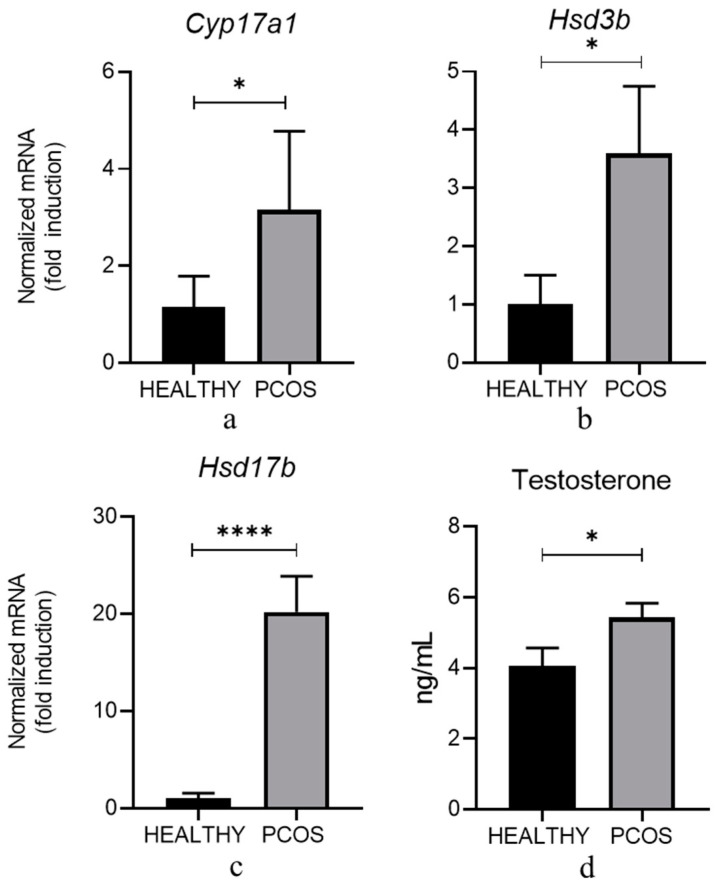
Expression levels of *Cyp17a1*, *Hsd3b*, and *Hsd17b* genes in ovaries from healthy and PCOS mice. Total RNA extracted from ovaries isolated from 10 healthy controls and 20 PCOS animals was subjected to real-time PCR using primers specific for (**a**) *Cyp17a1*, (**b**) *Hsd3b*, and (**c**) *Hsd17b* genes. Each sample was normalized to its *Rps29* content. Results are expressed as fold induction with respect to healthy mice arbitrarily set at 1 and are represented as the mean ± SEM. (**d**) Serum levels of Testosterone in the same healthy and PCOS mice as in panels (**a**–**c**). Data are presented as mean ± SEM and expressed as ng/mL. * *p* < 0.05; **** *p* < 0.0001 vs. healthy mice.

**Figure 3 cells-13-01171-f003:**
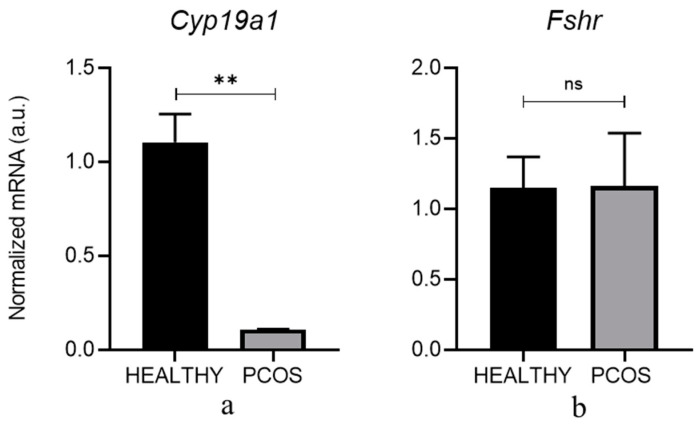
Basal expression levels of *Cyp19a1* and *Fshr* genes in ovaries from healthy and PCOS mice. Total RNA extracted from ovaries isolated from 10 healthy controls and 20 PCOS animals was subjected to real-time PCR using primers specific for (**a**) *Cyp19a1* and (**b**) *Fshr* genes. Each sample was normalized to its *Rps29* content. Results are expressed as arbitrary units (a.u.) and are represented as the mean ± SEM. ** *p* < 0.01, ns: not statistical.

**Figure 4 cells-13-01171-f004:**
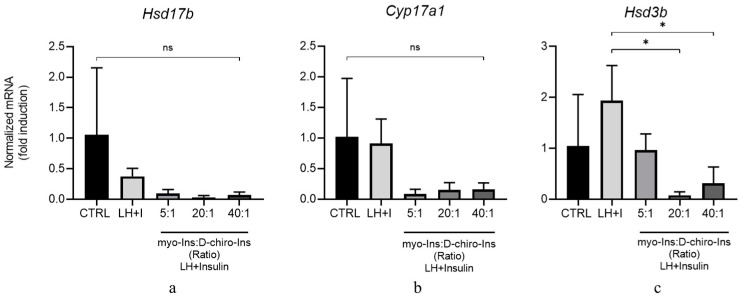
Inositol treatment of TCs with different myo-Ins and D-chiro-Ins ratio. TCs isolated from PCOS (*n* = 3 × 10) mice were pre-treated for 10 min with LH/Insulin (LH + I) and then treated for 90 min with myo-Ins/D-chiro-Ins at different ratios in the continuous presence of LH + Insulin. Total RNA was extracted at the end of the culture and subjected to real-time PCR using primers specific for (**a**) *Hsd17b*, (**b**) *Cyp17a1*, and (**c**) *Hsd3b* genes. Data are presented as the mean ± SEM of three independent experiments performed in triplicate and are expressed as fold induction with respect to the control arbitrarily set at 1. * *p* < 0.05, ns: not statistical.

**Figure 5 cells-13-01171-f005:**
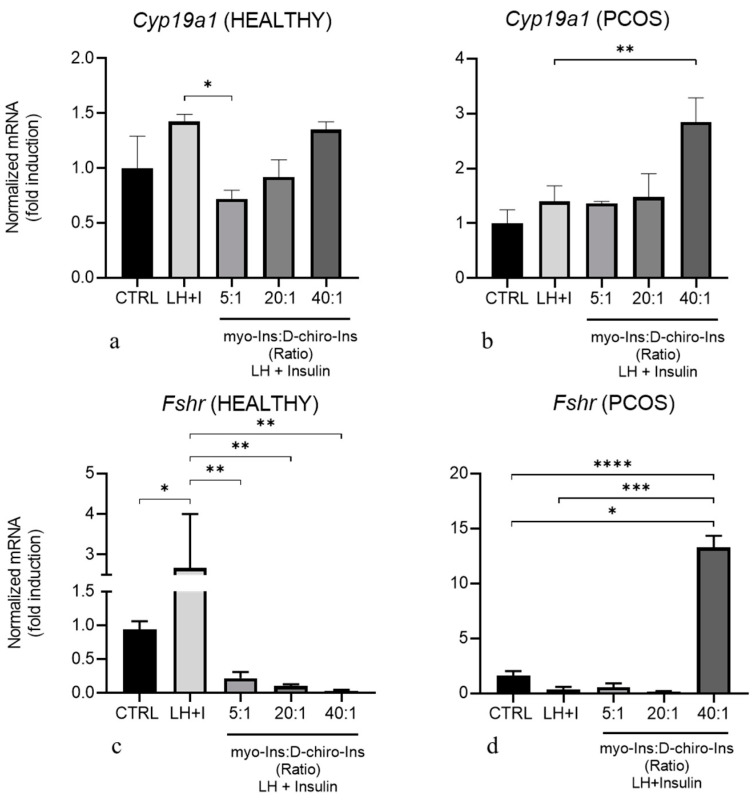
Inositol treatment of GCs at different pharmacological myo-Ins/D-chiro-Ins ratios. GCs isolated from healthy (*n* = 3 × 10) and PCOS (*n* = 3 × 10) mice were pre-treated for 10 min with LH/Insulin (LH + I) and then treated for 90 min with myo-Ins/D-chiro-Ins at different ratios in the continuous presence of LH + Insulin. Total RNA was extracted at the end of the culture and subjected to real-time PCR using primers specific for *Cyp19a1* and *Fshr* genes in GCs from Healthy (**a**,**c**) and PCOS (**b**,**d**) mice. Data are presented as mean ± SEM of three independent experiments performed in triplicate and are expressed as fold induction with respect to control arbitrarily set at 1. * *p* < 0.05, ** *p* < 0.01, *** *p* < 0.001, **** *p* < 0.0001.

**Figure 6 cells-13-01171-f006:**
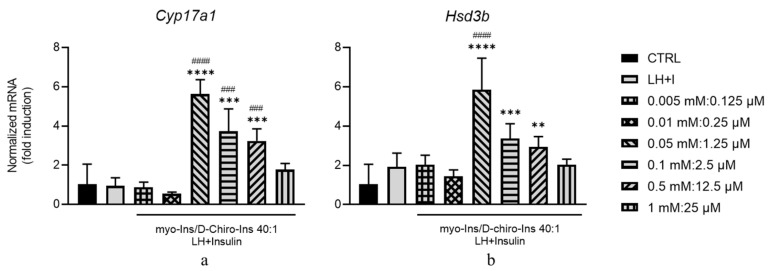
Inositol treatment of TCs at different physiological myo-Ins and D-chiro-Ins ratios. TCs isolated from PCOS mice (*n* = 3 × 16) were pre-treated for 10 min with LH/Insulin (LH + I) and then treated for 90 min, in the continuous presence of LH + Insulin with myo-Ins/D-chiro-Ins at different physiological concentrations shown in Table 3 and in the legend. Total RNA was extracted at the end of the culture and subjected to real-time PCR using primers specific for (**a**) *Cyp17a1* and (**b**) *Hsd3b* genes. Data are presented as the mean ± SEM of three independent experiments performed in triplicate and are expressed as fold induction with respect to control (CTRL) arbitrarily set at 1. ** *p* < 0.01, *** *p* < 0.001, **** *p* < 0.0001, vs., CTRL and ### *p* < 0.001, #### *p* < 0.0001) vs. LH + I.

**Figure 7 cells-13-01171-f007:**
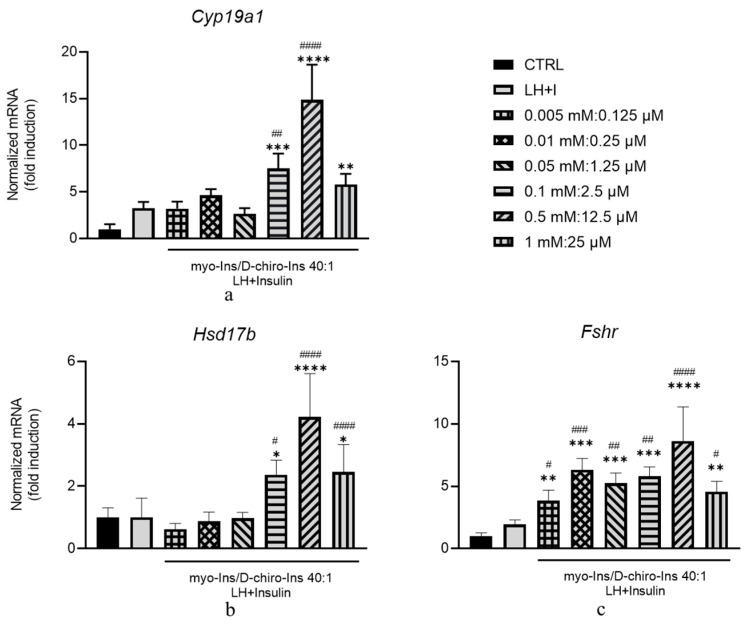
Inositol treatment of GCs at different physiological myo-Ins/D-chiro-Ins ratios. GCs isolated from PCOS mice were pre-treated for 10 min with LH/Insulin (LH + I) and then treated for 90 min in the continuous presence of LH + Insulin with myo-Ins and D-chiro-Ins at different physiological concentrations shown in Table 3 and in the legend. Total RNA was extracted at the end of the culture and subjected to real-time PCR using primers specific for (**a**) *Cyp17a1*, (**b**) *Hsd3b*, (**c**) and *Fshr* genes. Data are presented as mean the ± SEM of three independent experiments performed in triplicate and are expressed as fold induction with respect to CTRL arbitrarily set at 1. * *p* < 0.05, ** *p* < 0.01, *** *p* < 0.001, **** *p* < 0.0001 vs. CTRL and # *p* < 0.05, ## *p* < 0.01, ### *p* < 0.001, #### *p* < 0.0001) vs. LH + I.

**Figure 8 cells-13-01171-f008:**
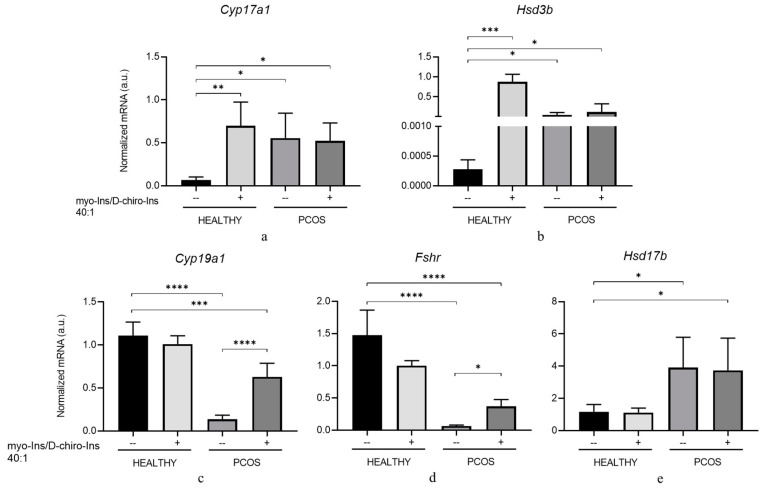
Effects of in vivo inositol treatment on the expression levels of steroidogenic enzymes in ovaries from healthy and PCOS mice. Ten healthy and 20 PCOS mice were in vivo treated with supra-physiological concentrations of myo-Ins and D-chiro-Ins (0.5 mM and 12.5 µM for myo-Ins and DCI, respectively) at a ratio of 40:1 for 10 days (+). Five healthy and five PCOS mice were not treated (--) and utilized as controls. At the end of treatment, ovaries were isolated, total RNA extracted, and subjected to real-time PCR using primers specific for (**a**) *Cyp17a1*, (**b**) *Hsd3b*, (**c**) *Cyp19a1*, (**d**) *Fshr*, and (**e**) *Hsd17b*. Results are expressed as arbitrary units (a.u.) and are represented as the mean ± SEM. * *p* < 0.05, ** *p* < 0.01, *** *p* < 0.001, **** *p* < 0.0001.

**Figure 9 cells-13-01171-f009:**
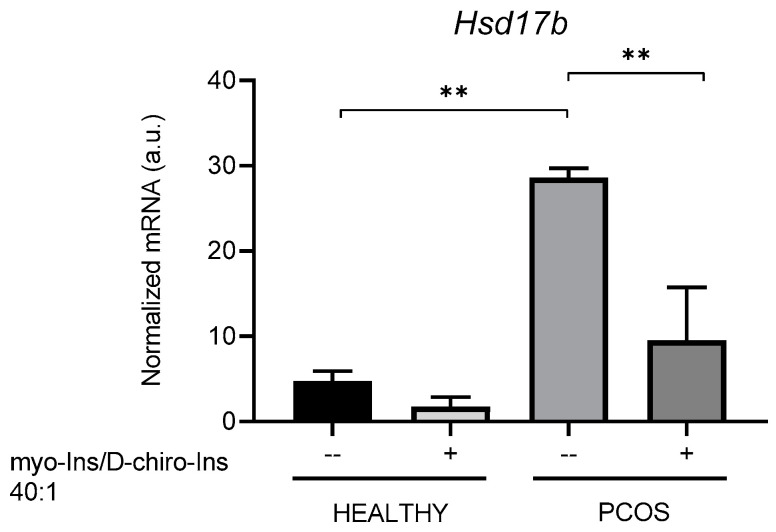
Expression of *Hsd17b* into adipose tissue after in vivo treatment with Inositols. Ten healthy and 20 PCOS mice were in vivo treated with supra-physiological concentrations of myo-Ins/D-chiro-Ins (0.5 mM and 12.5 µM for myo-Ins and DCI, respectively) at a ratio of 40:1 for 10 days (+). Five healthy and five PCOS mice were not treated (--) and utilized as controls. At the end of treatment, adipose tissue was isolated, total RNA extracted, and subjected to real-time PCR using primers specific for *Hsd17b*. Results are expressed as arbitrary units (a.u.) and are represented as the mean ± SEM. ** *p* < 0.01.

**Figure 10 cells-13-01171-f010:**
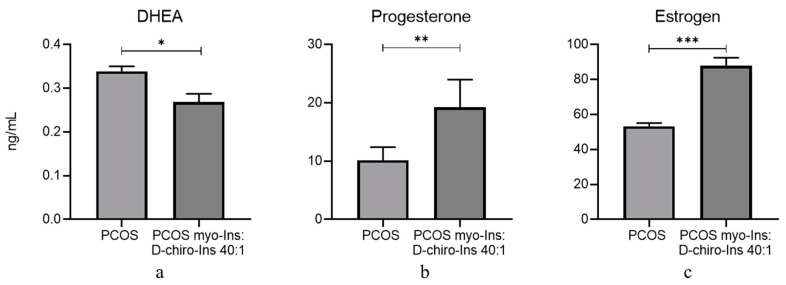
Hormone levels in the serum of PCOS mice treated and not treated with inositols. Ten healthy and 20 PCOS mice were in vivo treated with supra-physiological concentrations of myo-Ins and D-chiro-Ins (0.5 mM and 12.5 µM for myo-Ins and DCI, respectively) at a ratio of 40:1 for 10 days (+). Five healthy and five PCOS mice were not treated (--) and utilized as controls. At the end of treatment, blood was collected, and DHEA (**a**), progesterone (**b**), and estrogens (**c**) were evaluated by ELISA. Results are expressed as ng/mL and are represented as the mean ± SEM. * *p* < 0.05, ** *p* < 0.01, *** *p* < 0.001.

**Figure 11 cells-13-01171-f011:**
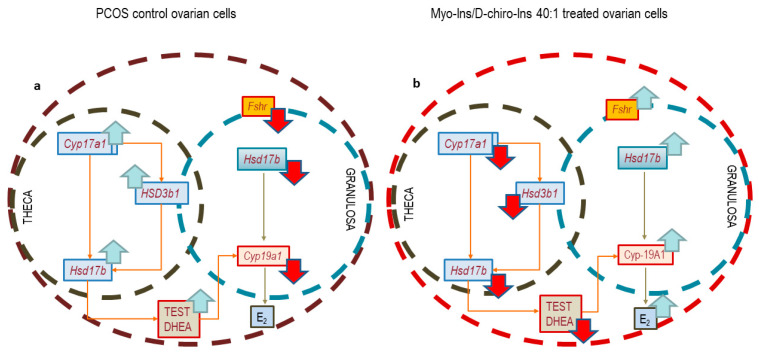
Schematic representation of the endocrine modulation triggered by myo-Ins/D-chiro-Ins in TCs and GCs. (**a**) TCs and GCs from PCOS ovary: androgenic enzyme (*Cyp17a1*, *Hsd3b1*, and *Hsd17b*) expression increases with respect to values found in healthy ovaries. Consequently, testosterone (TEST) and DHEA have significantly increased. We observe an impressive downregulation of *Hsd17b*, *Cyp19a1*, and *Fshr* in GCs, with resulting low levels of 17b-Estradiol. (**b**) The addition of myo-Ins/D-chiro-Ins (in the 40:1 ratio) induces a reversion of the overall picture: androgenic enzyme expression decreases, whereas *Fshr* and *Cyp19a1* significantly upsurge, with a resulting increase in estrogen availability (E_2_).

**Table 1 cells-13-01171-t001:** Sequence of oligonucleotides used as real-time PCR primers.

Gene	Sequence	
*Rps29*	FW	5′-TTCCTTTCTCCTCGTTGGGC-3′
	RV	5′-TTCAGCCCGTATTTGCGGAT-3′
*Cyp19a1*	FW	5′-GGATTGGAAGTGCCTGCAAC-3′
	RV	5′-CATGCTTGAGGACTTGCTGA-3′
*Fshr*	FW	5′-TGGGCCAGTCGTTTTAGACAT-3′
	RV	5′-AGGGAGCTTTTTCAAGCGGT-3′
*Hsd17b*	FW	5′-TGGACGTGCTGGTGTGTAAC-3′
	RV	5′-GTCCCCGTTAGGTTCACGTC-3′
*Cyp17a1*	FW	5′-GGAGAGTTTGCCATCCCGAA-3′
	RV	5′-TCTAAGAAGCGCTCAGGCAT-3′
*Hsd3b*	FW	5′-CGGCTGCTGCACAGGAATAA-3′
	RV	5′-ATGCCTGCTTCGTGACCATA-3′

FW, forward primer; RV, reverse primer.

**Table 2 cells-13-01171-t002:** Different ratios of myo-Ins and D-Chiro-Ins at pharmacological concentrations used for ovarian cell stimulation.

Myo-Ins	D-Chiro-Ins	Ratio
1 mM	200 µM	5:1
1 mM	50 µM	20:1
1 mM	25 µM	40:1

**Table 3 cells-13-01171-t003:** The table shows the concentrations of myo- and D-chiro-Ins used at a 40:1 ratio. The concentrations in bold are pharmacological concentrations. Supra-physiological concentrations are represented in the gray row.

Myo-Ins	D-Chiro-Ins	Ratio
0.005 mM	0.125 µM	40:1
0.01 mM	0.25 µM	40:1
0.05 mM	1.25 µM	40:1
0.1 mM	2.5 µM	40:1
0.5 mM	12.5 µM	40:1
**1 mM**	**25 µM**	**40:1**

## Data Availability

All data generated or analyzed during this study are included in this published article or in the data repositories listed in References.

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
