# Peer review of "Inositol Restores Appropriate Steroidogenesis in PCOS Ovaries Both In Vitro and In Vivo Experimental Mouse Models"

_cells, 2024, doi:10.3390/cells13141171_

Round 1

Reviewer 1 Report

Comments and Suggestions for Authors

The authors described about the inositol administration restoring the  steroidogenesis in PCOS ovaries both in vitro and in vivo experimental mouse models. The manuscript is well writing and interesting. Some points needs review.

1- In the introduction section, is important to add a clear hypothesis for the study.

2- In all methodology sections the experimental n must be added.

3- Line 123: DEHA, change for DHEA

4- It would be better to present the graphs as a scatter dot plot

5- I suggest adding a schematic figure to explain the mechanisms found in the study, taking into account the mechanisms that occur in the ovary of the experimental model without and with inositol treatment.

Comments on the Quality of English Language

Please, before the final version, review all the English in the text and the acronyms used.

Reviewer 2 Report

Comments and Suggestions for Authors

In the study reported in this manuscript, the authors investigated the role of myo-Inositol and D-chiro-Inositol on theca and granulosa cells from ovaries of CD1 female mice experimentally exposed to continuous light exposure for 10 weeks (70 days) as a model of PCOS induction. For the study several enzymes regulated by FSH and responsible for the production of androgens and their conversion to estrogens by the granulosa cells were assessed at the level of mRNA induction, both in the absence and in the presence of different mixture of myo-inositol and D-Chiro-Inositol. In particular, the attention of the authors focused on the activity of CYP19A1, due to the emerging evidence of a key role of cytochrome-based aromatase in the conversion of androgens to estrogens under normal conditions and under PCOS conditions.  The effect of treatment with different combinations of myo-Inositol/D-chiro-Inositol on the expression of FSHr, androgenic and estrogenic enzymes was analyzed by rt-PCR in isolated theca and granulosa cells isolated from healthy and PCOS mice. The results reported in the study indicate that a 40:1 ratio at pharmacological and physiological concentrations positively modulate the expression of CYP19A1 and FSHR genes in granulosa cells Similar effects were also observed in vivo, and mirrored the modification s observed in the serum levels of hormones in the treated animals. The final conclusion of the authors is that the use of myo-Inositol and D-Chiro-Inositol at the 40:1 ratio effectively rescued the PCOS animals from their pathological conditions, while downregulating androgen excess production. The authors also indicate in the manuscript that a similar beneficial effect was observed on the 17b-HSD from adipose tissue. 

Comments: The study is well conceived and properly conducted. The results reported in the study support the conclusion of the authors. Whether the approach used by the authors in this study will also work in humans remains to be determined. 

Minor comments: 

1. Do the authors investigated whether the administration of myo-inositol and D-chiro-Inositol changed the serum levels of inhibins and activins from the granulosa cells as a possible modulatory mechanism on the expression of the aromatase?

2. The decrease in serum DHEA levels reported in the study would suggest that the  administration of myo-inositol and D-chiro-Inositol may also affects the reticularis cells in the adrenal gland of the animals. Have the authors assessed the levels of aromatase and 17a-hydroxylase and 17,20 desmolase in those cells, and the circulating levels of 17-OH-progesterone?
